# The landscape for HIV pre-exposure prophylaxis during pregnancy and breastfeeding in Malawi and Zambia: A qualitative study

**Chifundo Zimba[1]\*, Suzanne Maman[2], Nora E. Rosenberg[2], Wilbroad Mutale[3], Oliver Mweemba[4], Wezzie Dunda[1], Twambilile Phanga[1], Kasapo F. Chibwe[4], Tulani Matenga[4], Kellie Freeborn[5], Leah Schrubbe[5], Bellington Vwalika[6], Benjamin H. Chi[5]**

**1** UNC Project-Malawi, Tidziwe Center, Lilongwe, Malawi, **2** Department of Health Behavior, Gillings School of Global Public Health, University of North Carolina at Chapel Hill, Chapel Hill, North Carolina, United States of America, **3** Department of Health Policy, School of Public Health, University of Zambia, Ridgeway Campus, Lusaka, Zambia, **4** Department of Health Promotion and Education, School of Public Health, University of Zambia, Ridgeway Campus, Lusaka, Zambia, **5** Department of Obstetrics and Gynecology, School of Medicine, University of North Carolina at Chapel Hill, Chapel Hill, North Carolina, United States of America, **6** Department of Obstetrics and Gynecology, School of Medicine, University of Zambia, Ridgeway Campus, Lusaka, Zambia

\* czimba@unclilongwe.org

**Data Availability Statement:** The data cannot be shared publicly due to ethical consideration surrounding this qualitative research. Specifically,

## Abstract

High HIV incidence rates have been observed among pregnant and breastfeeding women in sub-Saharan Africa. Oral pre-exposure prophylaxis (PrEP) can effectively reduce HIV acquisition in women during these periods; however, understanding of its acceptability and feasibility in antenatal and postpartum populations remains limited. To address this gap, we conducted in-depth interviews with 90 study participants in Malawi and Zambia: 39 HIV-negative pregnant/breastfeeding women, 14 male partners, 19 healthcare workers, and 18 policymakers. Inductive and deductive approaches were used to identify themes related to PrEP. As a public health intervention, PrEP was not well-known among patients and healthcare workers; however, when it was described to participants, most expressed positive views. Concerns about safety and adherence were raised, highlighting two critical areas for community outreach. The feasibility of introducing PrEP into antenatal services was also a concern, especially if introduced within already strained health systems. Support for PrEP varied among policymakers in Malawi and Zambia, reflecting the ongoing policy discussions in their respective countries. Implementing PrEP during the pregnancy and breastfeeding periods will require addressing barriers at the individual, facility, and policy levels. Multi-level approaches should be considered in the design of new PrEP programs for antenatal and postpartum populations.

interview transcripts may contain potential identifiable information and selected participants could be potentially identified via the context of the provided data. However, we will make these data available upon request for interested researchers. Such requests can be made by contacting Allison Gottwalt (allison_gottwalt@med.unc.edu).

**Funding:** This work received funding from the following sources: National Institutes of Health, R01 AI131060, Associated author(s): BHC, WM https://www.nih.gov. The funders had no role in study design, data collection and analysis, decision to publish, or preparation of the manuscript. National Institutes of Health, K24AI120796, Associated author(s): BHC, https://www.nih.gov. The funders had no role in study design, data collection and analysis, decision to publish, or preparation of the manuscript. National Institutes of Health, R00MH104154, Associated author(s): NER, https://www.nih.gov. The funders had no role in study design, data collection and analysis, decision to publish, or preparation of the manuscript. National Institutes of Health, P30AI050410, Associated author(s): BHC, https://www.nih.gov. The funders had no role in study design, data collection and analysis, decision to publish, or preparation of the manuscript. National Institutes of Health, D43 TW009340, Associated author(s): BHC, CZ, and KF, https://www.nih.gov. The funders had no role in study design, data collection and analysis, decision to publish, or preparation of the manuscript.

**Competing interests:** The authors have declared that no competing interests exist.

## Introduction

In many African settings, pregnancy and breastfeeding represent periods of elevated risk for HIV acquisition in women. Across 19 studies—and over 22,000 person-years of follow-up— Drake and colleagues found that HIV incidence rates during this period (3.8 per 100 person-years) exceeded World Health Organization incidence thresholds for at-risk populations (3.0 per 100 person-years) [1]. In 2018, the Joint United Nations Programme for HIV/AIDS (UNAIDS) estimated that 140,000 women newly acquired HIV during pregnancy and breast-feeding across 21 sub-Saharan African focus countries [2]. Although many have emphasized the importance of HIV prevention in antenatal settings [3–5], in most African settings, few HIV prevention modalities are available to pregnant and breastfeeding women.

Oral pre-exposure prophylaxis (PrEP) for HIV—in the form of co-formulated tenofovir (TDF) and emtricitabine (FTC)—has been studied extensively, shown to reduce HIV acquisition in women [6, 7], and to be safe for mothers and infants [8]. In PrEP studies among HIV-serodiscordant couples, early evidence on birth outcomes and infant growth has been reassuring [9–11]. PrEP during pregnancy and breastfeeding is also cost-effective in settings where HIV incidence is high [12]. In women, daily adherence is critical to its effectiveness. In studies where adherence was high, the risk for new HIV infections dropped by as much as 85% [6]; in studies demonstrating low adherence, results were equivocal [13].

Malawi and Zambia have generalized HIV epidemics, with both high fertility rates and high rates of HIV infection among women and children [14, 15]. Despite their geographic proximity, the underlying national policies for PrEP between these countries have differed. PrEP was first included as part of Zambia's national HIV guidelines beginning in 2016 and subsequent policies have emphasized its role for pregnant/breastfeeding women at elevated risk for acquiring HIV [15]. In contrast, the current Malawi national guidelines for HIV do not include PrEP as part of HIV prevention [16, 17]. However, PrEP has been incorporated into the country's latest national HIV prevention strategy (2018–2020), in particular its use among adolescent girls and young women [16]. In preparation for a pilot randomized trial of combination HIV prevention, we conducted formative qualitative research to further assess the landscape for PrEP during pregnancy/breastfeeding in these two countries. Specifically, we sought to evaluate the potential acceptability and feasibility of PrEP in pregnancy/breastfeeding in settings where—like much of sub-Saharan Africa—the adoption and implementation of PrEP policies has only recently begun.

## Materials and methods

This qualitative study was part of a larger effort to design and evaluate a combination HIV prevention package for pregnant and breastfeeding women and their partners [3].

This formative work was conducted in Malawi and Zambia, where the parent study will be implemented. In both countries, population-based surveys have reported high rates of annual HIV incidence among women of reproductive age (0.46%, 95%CI: 0.18–0.75% in Malawi and 1.10%, 95%CI: 0.72–1.48% in Zambia) [18, 19]. Although data are not available to describe population-level HIV incidence during pregnancy and breastfeeding, prior studies from Malawi and Zambia indicate substantially higher rates during these periods [20–23]. In addition, estimates from the 2018 UNAIDS Spectrum model suggest that new maternal HIV infections—acquired during pregnancy and breastfeeding—may contribute to as high as 40–45% of new infant infections in these countries [2].

We used a qualitative descriptive approach in the design, data collection and analysis of this formative study [24, 25]. Our goal was to provide accurate accounting of events—and their meaning—from the individuals interviewed [24]. We drew from the basic tenets of naturalistic

inquiry, with no specific commitment to a pre-defined theoretical framework [26]. We recruited participants from four populations: HIV-negative pregnant or breastfeeding women seeking care, their male partners, healthcare workers (HCWs), and policymakers. The first three groups were enrolled from Bwaila District Hospital (Lilongwe, Malawi), the University Teaching Hospital (Lusaka, Zambia), and Kamwala Health Centre (Lusaka, Zambia). Policymakers from both Malawi and Zambia were identified through existing governmental technical working groups.

Female participants were recruited from maternal and child health care units (i.e., antenatal, postnatal, family planning, and well child clinics) at these hospitals via convenience sampling. All participants were aged 18 years or above and resided near the study sites. The inclusion criteria for pregnant/breastfeeding women were: confirmed pregnancy or reported delivery with continued breastfeeding, access of maternal-child health care services at one of the study sites, documented HIV-negative status, and report of a male sexual partner. Male partners of participating pregnant or breastfeeding women were eligible for this study. These individuals were only recruited after we received permission from the index pregnant or breastfeeding participant. HCWs were eligible to participate if they worked at the maternal and child health care units within the targeted study health facilities. Policymakers were recruited from the Ministries of Health, national AIDS commissions, partner implementing organizations, and donor agencies. All participated on governmental HIV technical working groups. Our target accrual was: 40 HIV-negative pregnant/breastfeeding women, 40 male partners, 20 HCWs, and 20 policymakers. This sample size was divided equally between the two countries.

We conducted in-depth interviews to better understand the role of PrEP in HIV prevention among pregnant and breastfeeding women. Women, male partners, HCW, and policymakers were all asked about their knowledge of PrEP, overall opinion/acceptability of PrEP, and perceived challenges for implementing PrEP in pregnant/breastfeeding women in Malawi and Zambia. Sociodemographic data for pregnant/breastfeeding women and male partners were collected using a separate form. We did not collect analogous information from HCWs and policymakers for reasons of confidentiality.

Researchers and research assistants trained in qualitative methods conducted the in-depth interviews to capture participant views on PrEP use during pregnancy and breastfeeding. All interviewers were local; they spoke the language of the participants and were familiar with the local context and culture. They used semi-structured interview guides developed by members of the study team. Prior to implementation, these interview guides were field tested at participating health facilities to assess content and ensure meaning. All interviews with pregnant/breastfeeding women, male partners, and HCWs were conducted in private rooms within study site facilities. Policymakers were typically interviewed at a private venue of their choosing. Interviews were conducted either in English or in local languages (e.g., Chichewa, Nyanja, or Bemba) depending on the participant's choice. Each interview lasted approximately one hour.

All in-depth interviews were audio-recorded, transcribed verbatim, and translated into English by bilingual study personnel. These were then verified for clarity and completeness by an independent reviewer who was part of the qualitative team but not involved in transcription and translation. Members of the study team developed a central codebook that was used in both countries. We used NVivo12 Version 10 (QSR International, Pty Ltd.; Doncaster, Victoria, Australia) to organize and code data. Initial inductive codes were derived from the interview guides and additional deductive codes were added as themes emerged. This information was then used to create summaries and matrices to compare participant views.

**Table 1. In-depth interview participants stratified by group and by country (June 2017 to September 2018).**

|  | HIV-negative women | Male partners of HIV-negative women* | Healthcare workers | Policymakers |
|---|---|---|---|---|
| Malawi | 20 | 7 | 10 | 10 |
| Zambia | 19 | 7 | 9 | 8 |
| Total | 39 | 14 | 19 | 18 |

* Male partners recruited only with permission of the participating index HIV-negative women.

We received ethical approval from the University of North Carolina at Chapel Hill Institutional Review Board (Chapel Hill, NC, USA), the National Health Science Research Committee of Malawi (Lilongwe, Malawi), and the University of Zambia Biomedical Research Ethics Committee (Lusaka, Zambia) to conduct this study. All participants provided written informed consent prior to initiating study activities.

## Results

From June 2017 to September 2018, we recruited 90 individuals from Malawi and Zambia to participate in in-depth interviews. This included 39 HIV-negative pregnant/breastfeeding women, 14 male partners, 19 HCWs, and 18 policymakers. Table 1 displays a breakdown of the study participants by type and country of interview. Sociodemographic characteristics of the pregnant/breastfeeding women and their male partners are shown in Table 2. Analogous data were not collected from healthcare workers or policymakers. From these interviews, four topics were covered: (1) existing knowledge about PrEP, (2) opinions and perceived

**Table 2. Sociodemographic characteristics of participating HIV-negative women and their male partners (June 2017 to September 2018).**

| Characteristic | HIV-negative women (N = 39) | Male partners of HIV-negative women (N = 14) |
|---|---|---|
| Median age in years (interquartile range) | 25 (22–32) | 30 (27–35) |
| Highest education level completed |  |  |
| None | 9 | 0 |
| Primary school | 13 | 6 |
| Secondary school | 8 | 2 |
| Tertiary school | 9 | 6 |
| Marital status |  |  |
| Married | 36 | 14 |
| Single | 3 | 0 |
| Separated/Divorced | 0 | 0 |
| Employment status |  |  |
| Employed (full-time or part-time) | 15 | 13 |
| Student | 2 | 0 |
| Unemployed/Other | 22 | 1 |
| HIV status |  |  |
| Negative | 39 | 13 |
| Positive | 0 | 0 |
| Unknown | 0 | 1 |

acceptability of PrEP, (3) potential barriers to implementation, and (4) potential solutions to address these barriers. In the following subsections, we present combined results from Malawi and Zambia, but highlight areas where country-level differences were observed. The implementation barriers and solutions are also presented together, stratified by individual, facility, and policy factors.

### Knowledge about PrEP

At the health facility level, knowledge about PrEP among was low. Most pregnant/breastfeeding women (31 of 40), male partners (9 of 14), and HCWs (13 of 19) had no knowledge of PrEP prior to this study. Those who believed they had heard about PrEP often confused with post-exposure prophylaxis (PEP). In contrast, almost all policymakers had prior knowledge of PrEP and possessed familiarity with its purpose and use. They all knew that PrEP was one of the HIV prevention strategies being considered in settings of high HIV prevalence.

### Opinions and perceived acceptability of PrEP

In all interviews, PrEP was described by interviewers as a medication that—if taken regularly by individuals who are HIV-negative—is effective in preventing new HIV infection, is formulated in a single pill, and requires a daily administration. After this description, most pregnant/breastfeeding women, male partners, and HCWs generally had positive opinions about the intervention. Most perceived PrEP as a good method to protect against HIV acquisition especially for HIV-negative individuals in HIV serodiscordant partnerships, those with multiple sexual partners, and those who believed their partners had ancillary partners.

> "I would use PrEP because if I were to ask him to come with me for testing he might refuse for me. . . So I wouldn't know what he does elsewhere. Because for him to cheat on his wife with me is the same way that he would cheat on me with another woman."

> (HIV-negative woman, Zambia)

Respondents stated that they would accept PrEP to prevent HIV transmission to their unborn children. Most pregnant/breastfeeding women (31 of 39) said they would agree to using PrEP in order to protect themselves and their infants from contracting HIV. Most pregnant/breastfeeding women said they could take PrEP in a manner similar to iron supplementation or antimalarial prophylaxis, both of which are dispensed at antenatal clinics.

> "[PrEP is] good because if the man is infected, you don't get infected, even the child that you are expecting cannot be infected with the virus."

> (HIV-negative woman, Malawi)

Among those indicating a willingness to initiate PrEP, some said they would like more information before making a final decision. This included knowledge of their HIV risk, drug benefits, and potential side effects for both mother and infant. In both Malawi and Zambia, most pregnant/breastfeeding women preferred to discuss PrEP use with their male partners prior to initiation.

> "I think I would talk to my husband first; I think that's the correct answer that I can give you at the moment because if you just start taking them he would ask where you got the

authority to start doing that. These things have steps you know. So I think is best you talk to him that there is this drug and it's given to people who have been exposed to HIV. . ."

(HIV-negative woman, Malawi)

Overall, of the partners who participated in the study, a little more than half (9 of 14) would advise their HIV-negative female partners to take PrEP during pregnancy or breastfeeding periods.

"I would advise her to take [PrEP] whole-heartedly because the baby comes in contact with so many things in the womb, so you would find that you infect the baby. I would advise her to take it every day."

(Male partner, Zambia)

We observed country-level differences in perceived acceptability among Malawian and Zambian male partners. Most Malawian male partners said they would advise their partners to take PrEP and only one was unsure because of potential side effects to the fetus. However, there were mixed opinions about PrEP among Zambian male partners. Of the seven respondents, three would advise their female partners to initiate PrEP, three would not advise their female partners to use PrEP, and one was uncertain. Those who supported PrEP said they favored it to prevent HIV not only in their female partners but also to protect their infants. Male partners said PrEP would most benefit women whose male partners have unknown HIV status or are in an HIV discordant relationship. Men who did not support PrEP use expressed concerns about its potential side effects to the woman. They also thought PrEP could promote promiscuity in women, since the fear of new HIV infection would be removed as a potential deterrent to multiple partnerships.

Among policymakers, all seven from Zambia had positive opinions about PrEP, while most from Malawi (8 of 10) expressed greater uncertainty. Policymakers in Zambia were eager to implement the guidelines for PrEP which had already been evaluated in pilot programs within the country. In contrast, a major theme among Malawian policymakers was a need for more local implementation evidence before the intervention should be adopted as one of the HIV prevention strategies. Malawian policymakers were uncertain if health care systems would manage PrEP because there are no guidelines to screen women for PrEP and, for those who initiate, ensure their adherence over time:

"What are the screening tools in question, in terms of identifying this woman as high risk. . .And so of course screening pieces are really critical; without the screening pieces I don't think it's practical to have a PrEP program."

(Policymaker, Malawi)

Additionally, Malawian policymakers also had concerns about how the donor community would prioritize PrEP whether it would jeopardize current investments in HIV treatment.

"I think what would be interesting to see is, assuming we reach that stage where we have the evidence out there, how the donor community will react in terms of supporting those initiatives and in sustaining them. . .As a country like Malawi, how can we prioritize PrEP at the cost of the 700,000 people who are on ART and these 700,000 are all donor funded. Would the government want to think of PrEP for prevention and not just focus on those that are sick and require the drugs everyday?"

(Policy maker, Malawi)

## Individual-level implementation barriers and possible solutions

Overall, about one third of all respondents expressed concerns about the side effects of PrEP, including potential harm to the baby. Some pregnant/breastfeeding women and some male partners also worried about the need for strict adherence. They feared difficulties in remembering the daily dosing of PrEP could affect its effectiveness over time.

> "Since you're not feeling anything hurt, you just have to take it [PrEP] to protect yourself, so you might be forgetting sometimes."
>
> (HIV-negative woman, Zambia)

> "I don't think it is a good idea, maybe it depends on your risk factors but I don't think taking medication every day is a good idea. What is in the PrEP, are they antiretroviral drugs?. . ... If you are a high risk person then maybe, but it is very difficult for someone with low risk to take PrEP. . ... A lot of ART has side effects so how are you going to deal with the side effects?. . ... I wouldn't take it, even if I was at high risk, I wouldn't take it. Taking medication every day? It is hard to take prescribed medicine when you are sick, so what more when you are just fine?"
>
> (Health care worker, Malawi)

To address these barriers, participants from all groups emphasized the importance of sensitizing and educating the community at large about PrEP. This included information about its effectiveness and risk of side effects. Linking of PrEP services to couples' HIV counseling and testing was also viewed as a promising strategy to encourage male engagement, which in turn could increase PrEP uptake and adherence.

## Facility-level implementation barriers and possible solutions

Policymakers and HCWs expressed concerns about increased burden on health care systems if PrEP services were implemented or expanded. They emphasized the need for additional resources at the facility level to ensure that PrEP would be delivered effectively. This included additional funding for provider training (both clinically and operationally), appropriate screening tools for individual HIV risk assessment, effective approaches for promoting adherence, and PrEP-specific information, education, and communication materials for clinic attendees.

> "I think this package [PrEP program] to work well, you need to involve a lot of counsellors because if you involve a lot of nurses, it will fail because these nurses we work with,. . ... are doing a lot of work here at health centres. We are not doing one [type of] work, we are doing a lot of work (laughs) you can even see the way we are doing a lot of work. . ."
>
> (Health care worker, Zambia)

## Policy-level implementation barriers and possible solutions

All 8 policymakers in Zambia knew that PrEP was part of the national consolidated HIV guidelines. Most policymakers in Malawi (8 of 10) desired more local evidence about PrEP implementation to inform national policies and guidelines. Many were concerned about the feasibility of implementing such services, especially in the context of competing priorities and demands for HIV programs.

Other barriers identified by policymakers in both Malawi and Zambia included: (1) long-term funding and sustainability of PrEP, particularly when relying heavily on donor agencies, (2) the need for monitoring and evaluation tools, and (3) further development of clinical management guidelines, particularly for those who acquire HIV while on PrEP. At the time of data collection of this study, there were few clinical guidelines about how to identify high-risk patients in need of PrEP, how to start and stop PrEP, and how to manage and follow up a patient once PrEP is initiated.

"Where we have problems with PrEP is whether that intervention is sustainable. . .. It is a good intervention but how can we improve the health systems so that we can be able to monitor those women. We have to be monitoring their seroconversion, so I think that is the tricky part. . .how do we stretch ourselves that far, how would our health systems be able to manage such an intervention?"

(Policymaker, Malawi)

To address these policy-level barriers, policymakers emphasized the need for clinical guidelines and monitoring and evaluation tools to guide frontline providers in PrEP implementation. In Malawi, policymakers advocated for research to enrich the body of local evidence for PrEP. However, no explicit potential solutions were offered about long-term sustainability of PrEP programs in either Malawi or Zambia.

## Discussion

In this qualitative study, we found that knowledge of PrEP was low among patients and healthcare providers. When the PrEP intervention was described, most pregnant/breastfeeding women viewed it favorably and considered it an important component of HIV prevention during pregnancy/breastfeeding. Male partners in Zambia expressed mixed opinions about PrEP: about half would advise their female partners to initiate PrEP while the other half would not due to safety concerns or the perception that it would promote promiscuity in women. In contrast, nearly all policymakers had prior knowledge of PrEP but their opinions about the intervention differed by country. In Zambia, where PrEP policies have been in place since 2016, there was greater acceptance of its public health utility. In Malawi, policymakers expressed a desire for local clinical and implementation research before PrEP was considered at a national level. Across our participant groups, barriers were identified at the individual (e.g., knowledge and fear of side effects), facility (e.g., health system burdens), and policy (e.g., guidelines and tools, sustainability) levels.

Despite supportive guidance from the World Health Organization [27], the inclusion of PrEP in guidelines for pregnant/breastfeeding women across sub-Saharan Africa has been uneven. In some national programs (e.g., Zimbabwe, Nigeria), pregnant and breastfeeding women are recognized as high-risk groups that are eligible for PrEP. Other national HIV guidelines have taken a more cautious approach; these Ministries of Health await locally generated evidence about feasibility and acceptability before issuing formal recommendations [28]. Our study was conducted in countries where PrEP services were only recently incorporated into the national HIV strategic plan (Malawi) or remain in the early stages of public health implementation (Zambia). Neither national program has prioritized pregnant and breastfeeding women as a target for PrEP services, a policy context that mirrors much of sub-Saharan Africa. Our landscape analysis provides insights about the perceptions of PrEP in such settings and complements the growing body of research from pilot PrEP programs in antenatal settings [11, 29].

Given PrEP's recent introduction to policy discussions in Malawi and Zambia, the low levels of existing knowledge about PrEP among patients and HCWs was not surprising. This is similar to studies done in the United States where HIV at-risk women also lack knowledge of PrEP, despite its availability as an HIV prevention method [30, 31]. We were encouraged to find that, upon receiving basic information about PrEP, the majority of participants had positive opinions regarding the intervention. In fact, most pregnant and breastfeeding women stated that they would consider initiating PrEP if offered.

Despite these favorable opinions, the reported concerns among respondents require consideration, particularly in the design of future programs. For example, concerns about potential harm to mother and infant echo findings from other studies in the region [4, 32, 33]. Similar to other settings [30, 34], the need for near-perfect adherence raised concerns and participants worried about the consequences of HIV transmission in the context of poor PrEP adherence. As services scale up in Malawi and Zambia, community education and outreach will be critical to patient uptake of and adherence to PrEP in antenatal settings. Approaches that identify and address patient concerns—whether through peer/male engagement, support groups, or patient-centered counseling—will be essential to eliciting and addressing such problems. The underlying social and gender dynamics in these settings must also be considered. Male partner engagement represents a potential facilitator if the male partner is engaged and a barrier if he is not [35]. Interestingly, few respondents mentioned the potential for stigma associated with PrEP use, even though many pilot programs have targeted key populations at elevated HIV risk [36]. There was also little discussion about the potential for relationship conflict or social harms. These areas may require further consideration when designing community outreach activities for PrEP.

In Malawi and Zambia, the expansion of universal HIV treatment for pregnant and breastfeeding women (i.e., "Option B+") has greatly increased access to antiretroviral therapy among HIV-positive women. In 2018, UNAIDS reported that both country programs reached national coverage rates of 95% [2], an important milestone for HIV treatment programs globally [37]. Given the demands of establishing and maintaining such successful programs, concerns about the sustainability of new HIV prevention services—built upon the same antenatal platforms—were not surprising. Policymakers articulated a need for prioritization within HIV prevention and treatment services, emphasizing the need for sustainability in the context of donor funding.

Among HCWs and policymakers, many of the recommendations about PrEP were operational in nature. Respondents articulated the need for standardized screening and triage procedures for pregnant/breastfeeding women at elevated risk of HIV acquisition. This may include setting-specific algorithm based on age (e.g., adolescent girls and young women), medical history, partner HIV status, and/or at-risk behaviors [38]. They recommended detailed clinical management guidelines, including management of new HIV seroconverters as well as monitoring and evaluation tools. The design and implementation of sustainable models is critical and could have important downstream impacts on service uptake [39]. In Kenya, for example, the PrEP Implementation for Young Women and Adolescents project has integrated PrEP services within routine antenatal and postpartum services across 16 health facilities in Western Kenya. The additional time required of healthcare providers ranged from a median of 18 minutes (interquartile range [IQR]: 15–26) for those initiating PrEP and a median of 13 minutes (interquartile range [IQR]: 7–15) for each woman declining these services [40]. Integrated antiretroviral therapy and PrEP services—as have been implemented for serodiscordant couples [41]—could also be a promising strategy to embed PrEP into maternal and child health platforms. Task-shifting of certain tasks (e.g., eligibility screening, adherence counseling) to trained, lower-level health providers could

further streamline the provision of services and reduce additional strain on understaffed clinical units.

PrEP is an important tool for comprehensive HIV prevention [27], but its integration into existing national programs has varied. The responses from policymakers in Malawi and Zambia reflect these differences in policy discussions. In Malawi, government approval for the public implementation of PrEP was only recently articulated and remains limited to select populations [16]. The policy change occurred after the completion of our study; however, the concerns expressed in this study—and others similar to them—are likely to guide program implementation. This caution at the policy level likely reflects the country's resource-constrained and overburdened health system and its reluctance to initiate PrEP in this population without tools for effective risk stratification. In Zambia, policymakers were generally more supportive of PrEP services, reflecting PrEP's inclusion in the country's consolidated HIV guidelines [15]. Zambian policymakers more often stressed the importance of operationalizing these policy recommendations. This is in line with programmatic advances in Zambia, where PrEP demonstration projects are already in place for men who have sex with men, female sex workers, HIV serodiscordant couples and other key populations [42]. Despite our emphasis on pregnant/breastfeeding populations, there were few insights specific to these populations or to the antenatal care platform that support them.

An important strength of our study was its broad representation from Malawi and Zambia. To better understand the landscape for PrEP during pregnancy and breastfeeding periods, we solicited a diverse range of perspectives from stakeholders at different levels of the healthcare system. We acknowledge important limitations as well. First, the implementation of PrEP has only begun in our target countries, with little to no programmatic activity focused on pregnant/breastfeeding women. While policy and implementation context mirrors much of sub-Saharan Africa, our findings may be less relevant where PrEP programs have progressed more rapidly. Second, although PrEP was viewed favorably by participating pregnant/breastfeeding women and some male partners, we recognize that decision-making remained hypothetical. PrEP was not offered to participants following their in-depth interviews, so uptake and early adherence could not be assessed. Multiple studies have shown inconsistencies between intention and action, including for PrEP [43]. Finally, the qualitative approach we undertook provided greater depth of insights about HIV prevention during pregnancy and breastfeeding; however, we recognize that the responses may not be fully representative of target populations. Those who were recruited may have had stronger opinions about these topics than those who declined participation; however, this was not captured in our pre-enrollment period.

## Conclusion

In summary, we conducted a landscape analysis of PrEP for pregnant and breastfeeding women, the first of its kind in Malawi and Zambia. We report a diverse range of perspectives, suggesting that PrEP for pregnancy and breastfeeding may be acceptable and feasible, but key barriers must be addressed for optimal service delivery. These results can help to inform strategies to increase uptake and retention along the PrEP cascade, particularly in settings where PrEP policies are newly introduced.

## Author Contributions

**Conceptualization:** Chifundo Zimba, Suzanne Maman, Nora E. Rosenberg, Wilbroad Mutale, Oliver Mweemba, Bellington Vwalika, Benjamin H. Chi.

**Data curation:** Chifundo Zimba, Wezzie Dunda, Twambilile Phanga, Kasapo F. Chibwe, Tulani Matenga.

**Formal analysis:** Chifundo Zimba, Wezzie Dunda, Kellie Freeborn, Leah Schrubbe.

**Investigation:** Chifundo Zimba, Suzanne Maman, Nora E. Rosenberg, Wilbroad Mutale, Oliver Mweemba, Wezzie Dunda, Twambilile Phanga, Kasapo F. Chibwe, Tulani Matenga, Bellington Vwalika, Benjamin H. Chi.

**Methodology:** Suzanne Maman, Nora E. Rosenberg, Wilbroad Mutale, Oliver Mweemba, Bellington Vwalika, Benjamin H. Chi.

**Project administration:** Leah Schrubbe, Benjamin H. Chi.

**Supervision:** Benjamin H. Chi.

**Writing – original draft:** Chifundo Zimba, Benjamin H. Chi.

**Writing – review & editing:** Chifundo Zimba, Suzanne Maman, Nora E. Rosenberg, Wilbroad Mutale, Oliver Mweemba, Wezzie Dunda, Twambilile Phanga, Kasapo F. Chibwe, Tulani Matenga, Kellie Freeborn, Leah Schrubbe, Bellington Vwalika, Benjamin H. Chi.

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
