## [Decision Letter · Decision Letter 0]

5 Aug 2019

PONE-D-19-16981

The landscape for HIV pre-exposure prophylaxis during pregnancy and breastfeeding in Malawi and Zambia: a qualitative study

PLOS ONE

Dear Dr Chi,

Thank you for submitting your manuscript to PLOS ONE. After careful consideration, we feel that it has merit but does not fully meet PLOS ONE’s publication criteria as it currently stands. Therefore, we invite you to submit a revised version of the manuscript that addresses the points raised during the review process.

We would appreciate receiving your revised manuscript by Sep 19 2019 11:59PM. To enhance the reproducibility of your results, we recommend that if applicable you deposit your laboratory protocols in protocols.io, where a protocol can be assigned its own identifier (DOI) such that it can be cited independently in the future. For instructions see: http://journals.plos.org/plosone/s/submission-guidelines#loc-laboratory-protocols

We look forward to receiving your revised manuscript.

Kind regards,

Dhayendre Moodley

Academic Editor

PLOS ONE

Journal Requirements:

2. Please include a copy of the interview guide used in the study, in both the original language and English, as Supporting Information, or include a citation if it has been published previously.

Additional Editor Comments:

Congratulations on a well conducted study and mostly an unbiased account of your diverse findings. A few suggestions to strengthen the readership interest:

1. If available, include recent HIV incidence and MTCT rates in pregnant and breastfeeding populations in these two countries (an indication for prioritising this population in PrEP policies and guidelines).

2. In the Discussion, briefly discuss when Zambia and Malawi implemented the PMTCT programme, current status of the PMTCT programme, whether donor agencies primarily support the programme and your thoughts on how PrEP can be integrated into current PMTCT guidelines. Include discussions with HCW and policy makers if available.

3. Lastly, transcript for PM607 is repeated. You could either find another transcript or consolidate your interpretations/application while citing the transcript once.

Reviewers' comments:

Reviewer's Responses to Questions

**Comments to the Author**

1. Is the manuscript technically sound, and do the data support the conclusions?

Reviewer #1: Yes

2. Has the statistical analysis been performed appropriately and rigorously? 

Reviewer #1: N/A

3. Have the authors made all data underlying the findings in their manuscript fully available?

Reviewer #1: Yes

4. Is the manuscript presented in an intelligible fashion and written in standard English?

Reviewer #1: Yes

5. Review Comments to the Author

Reviewer #1: Thanks for the opportunity to review this manuscript. It is clearly written and interesting. Here

are my comments:

Introduction:

1) Line 54 “Policymakers were identified from existing governmental technical working groups”

Is this in both countries?

Methods:

2) Please include approximate length of the interviews.

3) Please include the approach for participant selection.

4) Could the authors describe a little more about the participant selection; was the

intention to interview 39 women? Was age or parity a criteria? Why 7 men? Please

give a little more detail.

5) Please include the theoretical approach for analysis.

Results:

1) Could you add the dates to the title in Tables 1& 2 ?

2) Were there any differences in sociodemographics by country?

3) Please remove the participant ID numbers. This is not helpful to the reader.

4) Do we know the HIV status of the male partners?

5) Would be nice to know the gender and age of the health care workers and policy

makers. Is this possible?

Discussion:

Would be nice to discuss how this could be similar or different in different populations. What

about younger women 15-18years? What about higher risk groups?

Is there any data about PrEP in other groups of adults in Zambia or Malawi?

As noted in the limitations – there are PrEP programs already in place in most countries now,

this is less interesting since its based on intention rather than on real experience.

6. PLOS authors have the option to publish the peer review history of their article (what does this mean?). If published, this will include your full peer review and any attached files.

Reviewer #1: No

---

## [Author Response · Author response to Decision Letter 0]

16 Sep 2019

RESPONSE TO REVIEWERS

Editor’s comments

Congratulations on a well conducted study and mostly an unbiased account of your diverse findings. A few suggestions to strengthen the readership interest:

1. If available, include recent HIV incidence and MTCT rates in pregnant and breastfeeding populations in these two countries (an indication for prioritising this population in PrEP policies and guidelines).

* This has been included in the revised manuscript (lines 51-59). We have included data about HIV incidence among women of reproductive age, based on population-based surveys in Malawi and Zambia. We also include data from historical studies of HIV incidence during pregnancy and breastfeeding in these countries. Finally, we provide estimates from UNAIDS about the relative contribution of new maternal HIV infections to new infant HIV infections in the two target countries. 

2. In the Discussion, briefly discuss when Zambia and Malawi implemented the PMTCT programme, current status of the PMTCT programme, whether donor agencies primarily support the programme and your thoughts on how PrEP can be integrated into current PMTCT guidelines. Include discussions with HCW and policy makers if available.

* Thank you for this suggestion. We have included a paragraph about the current status of the PMTCT programs in Malawi and Zambia, including their success in expanding antiretroviral therapy coverage. We discuss the concerns raised by policymakers about the prioritization between HIV prevention (PrEP) and treatment (antiretroviral therapy) in the context of donor-funded programs (lines 329-336). In the following paragraph (lines 338-353), we provide suggestions on how national HIV programs might integrate PrEP into existing maternal and child health platforms, including PMTCT. 

3. Lastly, transcript for PM607 is repeated. You could either find another transcript or consolidate your interpretations/application while citing the transcript once.

* Thank you for flagging this repetition. Transcript PM607 is now only cited once.

Reviewer #1

Thanks for the opportunity to review this manuscript. It is clearly written and interesting. Here

are my comments:

Introduction:

1) Line 54 “Policymakers were identified from existing governmental technical working groups”

Is this in both countries? 

* Yes. We have added a phrase to indicate that this applies to both countries

Methods:

1) Please include approximate length of the interviews. 

* Interviews lasted approximately one hour (line 100).

2) Please include the approach for participant selection.

* Patients and their partners were recruited via a convenience sample. The eligibility and approach are described further in lines 71-81. 

3) Could the authors describe a little more about the participant selection; was the intention to interview 39 women? Was age or parity a criterion? Why 7 men? Please

give a little more detail. 

* The eligibility criteria and approach are described in lines 71-81. Our target accrual was 40 HIV-negative pregnant or breastfeeding women, up to 40 primary male partners (based on the index women’s permission to recruit), 20 HCWs, and 20 policymakers. The sample was divided evenly across the two country sites (lines 81-83). 

4) Please include the theoretical approach for analysis.

* We have included additional text in the revised manuscript. In lines 61-64, it reads: “We used a qualitative descriptive approach in the design, data collection and analysis of this formative study [18, 19]. Our goal was to provide accurate accounting of events—and their meaning—from the individuals interviewed [18]. We drew from the basic tenets of naturalistic inquiry, with no specific commitment to a pre-defined theoretical framework [20].”

Results:

1) Could you add the dates to the title in Tables 1 & 2?

* This has been added to the titles for Tables 1 and 2.

2) Were there any differences in sociodemographics by country? 

* Because of our small sample size, we did not further stratify the participating women and their male partners by country. We were concerned that, with 20 or fewer participants in any one category, differences would be difficult to interpret. 

3) Please remove the participant ID numbers. This is not helpful to the reader. 

* These have been removed as requested.

4) Do we know the HIV status of the male partners? 

* Out of 14 male partners, 13 were reported to be HIV-negative. Only 1 partner had unknown HIV status (Table 2).

5) Would be nice to know the gender and age of the health care workers and policy

makers. Is this possible?

* We agree that this information could be helpful. Unfortunately, we did not collect this information for healthcare workers and policy makers for reasons of confidentiality. Because these groups are small, there were concerns that such details could lead in accidental disclosure. This is now explained in lines 89-90.

Discussion:

1) Would be nice to discuss how this could be similar or different in different populations. What

about younger women 15-18years? What about higher risk groups?

* In lines 340-342, we discuss the importance of screening and triage procedures for pregnant/breastfeeding women at elevated risk for HIV acquisition. Currently, there are few validated measures; however, it is acknowledged—including by respondents in this study—that such instruments could help to make PrEP services more efficient and sustainable. 

2) Is there any data about PrEP in other groups of adults in Zambia or Malawi? 

* Data about PrEP use in these countries remains limited. In Malawi, programs are only now beginning. In Zambia, pilot programs have been implemented for other key populations. These are now referenced in lines 365-367. 

3) As noted in the limitations – there are PrEP programs already in place in most countries now, this is less interesting since its based on intention rather than on real experience. 

* As noted in this comment, this limitation has been acknowledged within the paper. However, since the majority of countries have not implemented large-scale programs targeting pregnant and breastfeeding women specifically, we believe this remains a worthy contribution to the public health literature.

---

## [Editor Report · Decision Letter 1]

24 Sep 2019

The landscape for HIV pre-exposure prophylaxis during pregnancy and breastfeeding in Malawi and Zambia: a qualitative study

PONE-D-19-16981R1

Dear Dr. Chi,

We are pleased to inform you that your manuscript has been judged scientifically suitable for publication and will be formally accepted for publication once it complies with all outstanding technical requirements.

With kind regards,

Dhayendre Moodley

Academic Editor

PLOS ONE
---

## [Editor Report · Acceptance letter]

27 Sep 2019

PONE-D-19-16981R1 

The landscape for HIV pre-exposure prophylaxis during pregnancy and breastfeeding in Malawi and Zambia: a qualitative study 

Dear Dr. Chi:

I am pleased to inform you that your manuscript has been deemed suitable for publication in PLOS ONE. Congratulations! Your manuscript is now with our production department. 

With kind regards,

on behalf of

Prof. Dhayendre Moodley 

Academic Editor

PLOS ONE